# Continual Learning with Deep Generative Replay

**Hanul Shin**
Massachusetts Institute of Technology
SK T-Brain
skyshin@mit.edu

**Jung Kwon Lee**,* **Jaehong Kim**,* **Jiwon Kim**
SK T-Brain
{jklee,xhark,jk}@sktbrain.com

## Abstract

Attempts to train a comprehensive artificial intelligence capable of solving multiple tasks have been impeded by a chronic problem called catastrophic forgetting. Although simply replaying all previous data alleviates the problem, it requires large memory and even worse, often infeasible in real world applications where the access to past data is limited. Inspired by the generative nature of the hippocampus as a short-term memory system in primate brain, we propose the Deep Generative Replay, a novel framework with a cooperative dual model architecture consisting of a deep generative model ("generator") and a task solving model ("solver"). With only these two models, training data for previous tasks can easily be sampled and interleaved with those for a new task. We test our methods in several sequential learning settings involving image classification tasks.

## 1   Introduction

One distinctive ability of humans and large primates is to continually learn new skills and accumulate knowledge throughout the lifetime [6]. Even in small vertebrates such as rodents, established connections between neurons seem to last more than an year [13]. Besides, primates incorporate new information and expand their cognitive abilities without seriously perturbing past memories. This flexible memory system results from a good balance between synaptic plasticity and stability [1].

Continual learning in deep neural networks, however, suffers from a phenomenon called *catastrophic forgetting* [22], in which a model's performance on previously learned tasks abruptly degrades when trained for a new task. In artificial neural networks, inputs coincide with the outputs by implicit parametric representation. Therefore training them towards a new objective can cause almost complete forgetting of former knowledge. Such problem has been a key obstacle to continual learning for deep neural network through sequential training on multiple tasks.

Previous attempts to alleviate catastrophic forgetting often relied on episodic memory system that stores past data [31]. In particular, recorded examples are regularly replayed with real samples drawn from the new task, and the network parameters are jointly optimized. While a network trained in this manner performs as well as separate networks trained solely on each task [29], a major drawback of memory-based approach is that it requires large working memory to store and replay past inputs. Moreover, such data storage and replay may not be viable in some real-world situations.

Notably, humans and large primates learn new knowledge even from limited experiences and still retain past memories. While several biological mechanisms contribute to this at multiple levels, the most apparent distinction between primate brains and artificial neural networks is the existence of separate, interacting memory systems [26]. The Complementary Learning Systems (CLS) theory illustrates the significance of dual memory systems involving the hippocampus and the neocortex. The hippocampal system rapidly encodes recent experiences, and the memory trace that lasts for

---

a short period is reactivated during sleep or conscious and unconscious recall [8]. The memory is consolidated in the neocortex through the activation synchronized with multiple replays of the encoded experience [27]–a mechanism which inspired the use of experience replay [23] in training reinforcement learning agents.

Recent evidence suggests that the hippocampus is more than a simple experience replay buffer. Reactivation of the memory traces yields rather flexible outcomes. Altering the reactivation causes a defect in consolidated memory [35], while co-stimulating certain memory traces in the hippocampus creates a false memory that was never experienced [28]. These properties suggest that the hippocampus is better paralleled with a generative model than a replay buffer. Specifically, deep generative models such as deep Boltzmann machines [32] or a variational autoencoder [17] can generate high-dimensional samples that closely match observed inputs.

We now propose an alternative approach to sequentially train deep neural networks without referring to past data. In our deep generative replay framework, the model retains previously acquired knowledge by the concurrent replay of generated pseudo-data. In particular, we train a deep generative model in the generative adversarial networks (GANs) framework [10] to mimic past data. Generated data are then paired with corresponding response from the past task solver to represent old tasks. Called the scholar model, the generator-solver pair can produce fake data and desired target pairs as much as needed, and when presented with a new task, these produced pairs are interleaved with new data to update the generator and solver networks. Thus, a scholar model can both learn the new task without forgetting its own knowledge and teach other models with generated input-target pairs, even when the network configuration is different.

As deep generative replay supported by the scholar network retains the knowledge without revisiting actual past data, this framework can be employed to various practical situation involving privacy issues. Recent advances on training generative adversarial networks suggest that the trained models can reconstruct real data distribution in a wide range of domains. Although we tested our models on image classification tasks, our model can be applied to any task as long as the trained generator reliably reproduces the input space.

## 2   Related Works

The term *catastrophic forgetting* or *catastrophic interference* was first introduced by McCloskey and Cohen in 1980's [22]. They claimed that catastrophic interference is a fundamental limitation of neural networks and a downside of its high generalization ability. While the cause of catastrophic forgetting has not been studied analytically, it is known that the neural networks parameterize the internal features of inputs, and training the networks on new samples causes alteration in already established representations. Several works illustrate empirical consequences in sequential learning settings [7, 29], and provide a few primitive solutions [16, 30] such as replaying all previous data.

### 2.1   Comparable methods

A branch of works assumes a particular situation where access to previous data is limited to the current task[12, 18, 20]. These works focus on optimizing network parameters while minimizing alterations to already consolidated weights. It is suggested that regularization methods such as dropout [33] and L2 regularization help reduce interference of new learning [12]. Furthermore, elastic weight consolidation (EWC) proposed in [18] demonstrates that protecting certain weights based on their importance to the previous tasks tempers the performance loss.

Other attempts to sequentially train a deep neural network capable of solving multiple tasks reduce catastrophic interference by augmenting the networks with task-specific parameters. In general, layers close to inputs are shared to capture universal features, and independent output layers produce task-specific outputs. Although separate output layers are free of interference, alteration on earlier layers still causes some performance loss on older tasks. Lowering learning rates on some parameters is also known to reduce forgetting [9]. A recently proposed method called Learning without Forgetting (LwF) [21] addresses the problem of sequential learning in image classification tasks while minimizing alteration on shared network parameters. In this framework, the network's response to new task input prior to fine-tuning indirectly represents knowledge about old tasks and is maintained throughout the learning process.

## 2.2 Complementary Learning System(CLS) theory

A handful of works are devoted to designing a complementary networks architecture to alleviate catastrophic forgetting. When the training data for previous tasks are not accessible, only pseudo-inputs and pseudo-targets produced by a memory network can be fed into the task network. Called a pseudorehearsal technique, this method is claimed to maintain old input-output patterns without accessing real data [31]. When the tasks are as elementary as coupling two binary patterns, simply feeding random noises and corresponding responses suffices [2]. A more recent work proposes an architecture that resembles the structure of the hippocampus to facilitate continual learning for more complex data such as small binary pixel images [15]. However, none of them demonstrates scalability to high-dimensional inputs similar to those appear in real world due to the difficulty of generating meaningful high-dimensional pseudoinputs without further supervision.

Our generative replay framework differs from aforementioned pseudorehearsal techniques in that the fake inputs are generated from learned past input distribution. Generative replay has several advantages over other approaches because the network is jointly optimized using an ensemble of generated past data and real current data. The performance is therefore equivalent to joint training on accumulated real data as long as the generator recovers the input distribution. The idea of generative replay also appears in Mocanu et al. [24], in which they trained Restricted Boltzmann Machine to recover past input distribution.

## 2.3 Deep Generative Models

Generative model refers to any model that generates observable samples. Specifically, we consider deep generative models based on deep neural networks that maximize the likelihood of generated samples being in given real distribution [11]. Some deep generative models such as variational autoencoders [17] and the GANs [10] are able to mimic complex samples like images.

The GANs framework defines a zero-sum game between a generator $G$ and a discriminator $D$. While the discriminator learns to distinguish between the generated samples from real samples by comparing two data distributions, the generator learns to mimic the real distribution as closely as possible. The objective of two networks is thereby defined as:

$$\min_G \max_D V(D, G) = \mathbb{E}_{\boldsymbol{x} \sim p_{data}(\boldsymbol{x})}[\log D(\boldsymbol{x})] + \mathbb{E}_{\boldsymbol{z} \sim p_z(\boldsymbol{z})}[\log(1 - D(G(\boldsymbol{z})))]$$

# 3 Generative Replay

We first define several terminologies. In our continual learning framework, we define the sequence of tasks to be solved as a *task sequence* $\mathbf{T} = (T_1, T_2, \cdots, T_N)$ of $N$ tasks.

**Definition 1** *A task $T_i$ is to optimize a model towards an objective on data distribution $D_i$, from which the training examples $(\boldsymbol{x}_i, \boldsymbol{y}_i)$'s are drawn.*

Next, we call our model a *scholar*, as it is capable of learning a new task and teaching its knowledge to other networks. Note that the term scholar differs from standard notion of teacher-student framework of ensemble models [5], in which the networks either teach or learn only.

**Definition 2** *A scholar $H$ is a tuple $\langle G, S \rangle$, where a generator $G$ is a generative model that produces real-like samples and a solver $S$ is a task solving model parameterized by $\theta$.*

The solver has to perform all tasks in the task sequence $\mathbf{T}$. The full objective is thereby given as to minimize the unbiased sum of losses among all tasks in the task sequence $\mathbb{E}_{(\boldsymbol{x}, \boldsymbol{y}) \sim D}[L(S(\boldsymbol{x}; \theta), \boldsymbol{y})]$, where $D$ is the entire data distribution and $L$ is a loss function. While being trained for task $T_i$, the model is fed with samples drawn from $D_i$.

## 3.1 Proposed Method

We consider sequential training on our scholar model. However, training a single scholar model while referring to the recent copy of the network is equivalent to training a sequence of scholar models $(H_i)_{i=1}^N$ where the $n$-th scholar $H_n$ $(n > 1)$ learns the current task $T_n$ and the knowledge of previous scholar $H_{n-1}$. Therefore, we describe our full training procedure as in Figure 1(a).

Training the scholar model from another scholar involves two independent procedures of training the generator and the solver. First, the new generator receives current task input $x$ and replayed inputs $x'$ from previous tasks. Real and replayed samples are mixed at a ratio that depends on the desired importance of a new task compared to the older tasks. The generator learns to reconstruct cumulative input space, and the new solver is trained to couple the inputs and targets drawn from the same mix of real and replayed data. Here, the replayed target is past solver's response to replayed input. Formally, the loss function of the $i$-th solver is given as

$$L_{train}(\theta_i) = r\mathbb{E}_{(x,y)\sim D_i}[L(S(x;\theta_i),y)] + (1-r)\,\mathbb{E}_{x'\sim G_{i-1}}[L(S(x';\theta_i),S(x';\theta_{i-1}))] \quad (1)$$

where $\theta_i$ are network parameters of the $i$-th scholar and $r$ is a ratio of mixing real data. As we aim to evaluate the model on original tasks, test loss differs from the training loss:

$$L_{test}(\theta_i) = r\mathbb{E}_{(x,y)\sim D_i}[L(S(x;\theta_i),y)] + (1-r)\,\mathbb{E}_{(x,y)\sim D_{past}}[L(S(x;\theta_i),y)] \quad (2)$$

where $D_{past}$ is a cumulative distribution of past data. Second loss term is ignored in both functions when $i = 1$ because there is no replayed data to refer to for the first solver.

We build our scholar model with a solver that has suitable architecture for solving a task sequence and a generator trained in the generative adversarial networks framework. However, our framework can employ any deep generative model as a generator.

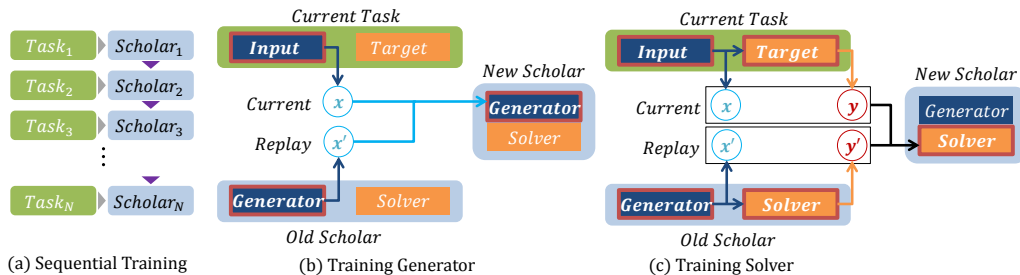

(a) Sequential Training    (b) Training Generator    (c) Training Solver

Figure 1: Sequential training of scholar models. (a) Training a sequence of scholar models is equivalent to continuous training of a single scholar while referring to its most recent copy. (b) A new generator is trained to mimic a mixed data distribution of real samples $x$ and replayed inputs $x'$ from previous generator. (c) A new solver learns from real input-target pairs $(x, y)$ and replayed input-target pairs $(x', y')$, where replayed response $y'$ is obtained by feeding generated inputs into previous solver.

## 3.2   Preliminary Experiment

Prior to our main experiments, we show that the trained scholar model alone suffices to train an empty network. We tested our model on classifying MNIST handwritten digit database [19]. Sequence of scholar models were trained from scratch through generative replay from previous scholar. The accuracy on classifying full test data is shown in Table 1. We observed that the scholar model transfers knowledge without losing information.

Table 1: Test accuracy of sequentially learned solver measured on full test data from MNIST database. The first solver learned from real data, and subsequent solvers learned from previous scholar networks.

| | $Solver_1 \rightarrow$ | $Solver_2 \rightarrow$ | $Solver_3 \rightarrow$ | $Solver_4 \rightarrow$ | $Solver_5$ |
|---|---|---|---|---|---|
| Accuracy(%) | 98.81% | 98.64% | 98.58% | 98.53% | 98.56% |

## 4   Experiments

In this section, we show the applicability of generative replay framework on various sequential learning settings. Generative replay based on a trained scholar network is superior to other continual learning approaches in that the quality of the generative model is the only constraint of the task performance. In other words, training the networks with generative replay is equivalent to joint training on entire data when the generative model is optimal. To draw the best possible result, we used WGAN-GP [14] technique in training the generator.

As a base experiment, we test if generative replay enables sequential learning while compromising performance on neither the old tasks nor a new task. In section 4.1, we sequentially train the networks on independent tasks to examine the extent of forgetting. In section 4.2, we train the networks on two different yet related domains. We demonstrate that generative replay not only enables continual learning on our design of the scholar network but also compatible with other known structures. In section 4.3, we show that our scholar network can gather knowledge from different tasks to perform a meta-task, by training the network on disjoint subsets of training data.

We compare the performance of the solver trained with variants of replay methods. Our model with generative replay is denoted in the figure as *GR*. We specify the upper bound by assuming a situation when the generator is perfect. Therefore, we replayed actual past data paired with the predicted targets from the old solver network. We denote this case as *ER* for exact replay. We also consider the opposite case when the generated samples do not resemble the real distribution at all. Such case is denoted as *Noise*. A baseline of naively trained solver network is denoted as *None*. We use the same notation throughout this section.

## 4.1   Learning independent tasks

The most common experimental formulation used in continual learning literature [34, 18] is a simple image classification problem where the inputs are images from MNIST handwritten digit database [19], but pixel values of inputs are shuffled by a random permutation sequence unique to each task. The solver has to classify permuted inputs into the original classes. Since the most, if not all pixels are switched between the tasks, the tasks are technically independent from each other, being a good measure of memory retention strength of a network.

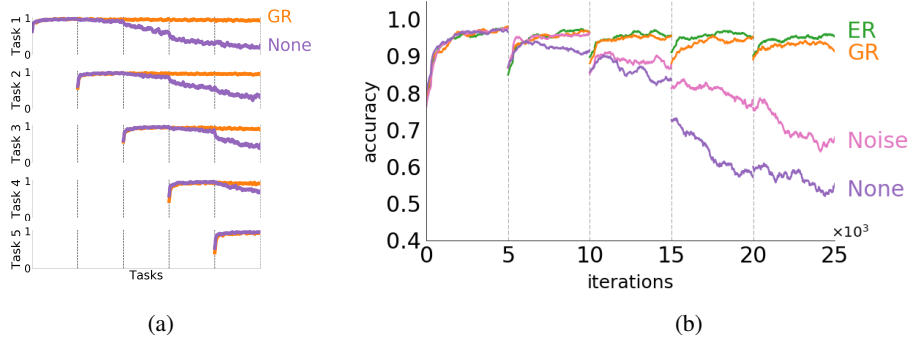

(a)                                                    (b)

Figure 2: Results on MNIST pixel permutation tasks. (a) Test performances on each task during sequential training. Performances for previous tasks dropped without replaying real or meaningful fake data. (b) Average test accuracy on learnt tasks. Higher accuracy is achieved when the replayed inputs better resembled real data.

We observed that generative replay maintains past knowledge by recalling former task data. In Figure 2(a), the solver with generative replay (orange) maintained the former task performances throughout sequential training on multiple tasks, in contrast to the naively trained solver (violet). An average accuracy measured on cumulative tasks is illustrated in Figure 2(b). While the solver with generative replay achieved almost full performance on trained tasks, sequential training on a solver alone incurred catastrophic forgetting (violet). Replaying random gaussian noises paired with recorded responses did not help tempering performance loss (pink).

## 4.2   Learning new domains

Training independent tasks on the same network is inefficient because no information is to be shared. We thus demonstrate the merit of our model in more reasonable settings where the model benefits from solving multiple tasks.

A model operating in multiple domains has several advantages over a model that only works in a single domain. First, the knowledge of one domain can help better and faster understanding of other domains if the domains are not completely independent. Second, generalization over multiple domains may result in more universal knowledge that is applicable to unseen domains. Such phenomenon is

also observed in infants learning to categorize objects [3, 4]. Encountering similar but diverse objects, young children can infer the properties shared within the category, and can make a guess of which category that the new object may belong to.

We tested if the model can incorporate the knowledge of a new domain with generative replay. In particular, we sequentially trained our model on classifying MNIST and Street View House Number (SVHN) dataset [25], and vice versa. Experimental details are provided in supplementary materials.

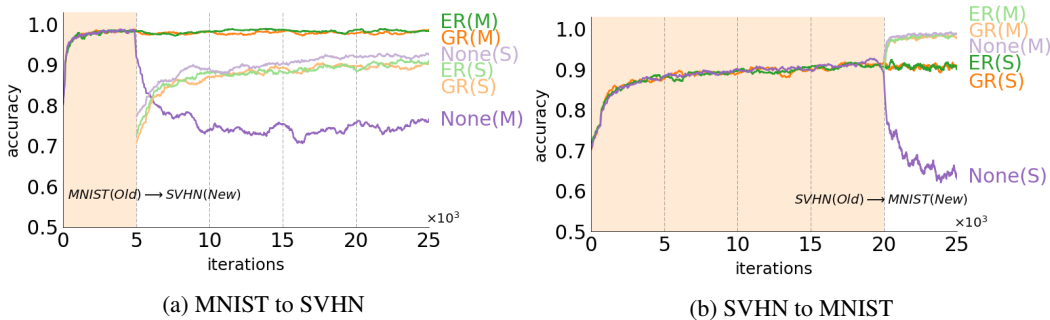

(a) MNIST to SVHN                    (b) SVHN to MNIST

Figure 3: Accuracy on classifying samples from two different domains. (a) The models are trained on MNIST then on SVHN dataset or (b) vice versa. When the previous data are recalled by generative replay (orange), knowledge of the first domain is retained as if the real inputs with predicted responses are replayed (green). Sequential training on the solver alone incurs forgetting on the former domain, thereby resulting in low average performance (violet).

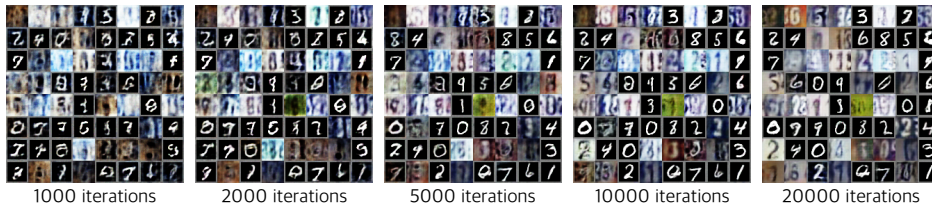

1000 iterations    2000 iterations    5000 iterations    10000 iterations    20000 iterations

Figure 4: Samples from trained generator in MNIST to SVHN experiment after training on SVHN dataset for 1000, 2000, 5000, 10000, and 20000 iterations. The samples are diverted into ones that mimic either SVHN or MNIST input images.

Figure 3 illustrates the performance on the original task (thick curves) and the new task (dim curves). A solver trained alone lost its performance on the old task when no data are replayed (purple). Since MNIST and SVHN input data share similar spatial structure, the performance on the former task did not drop to zero, yet the decline was critical. In contrast, the solver with generative replay (orange) maintained its performance on the first task while accomplishing the second one. The results were no worse than replaying past real inputs paired with predicted responses from the old solver (green). In both cases, the model trained without any replay data achieved slightly better performance on new task, as the network was solely optimized to solve the second task.

Generative replay is compatible with other continual learning models as well. For instance, Learning without Forgetting (LwF), which replays current task inputs to revoke past knowledge, can be augmented with generative models that produce samples similar to former task inputs. Because LwF requires the context information of which task is being performed to use task-specific output layers, we tested the performance separately on each task. Note that our scholar model with generative replay does not need the task context.

In Figure 5, we compare the performance of LwF algorithm with a variant LwF-GR, where the task-specific generated inputs are fed to maintain older network responses. We used the same training regime as proposed in the original literature, namely warming up the new network head for some amount of the time and then fine tuning the whole network. The solver trained with original LwF algorithm loses performance on the first task when fine-tuning begins, due to alteration to shared

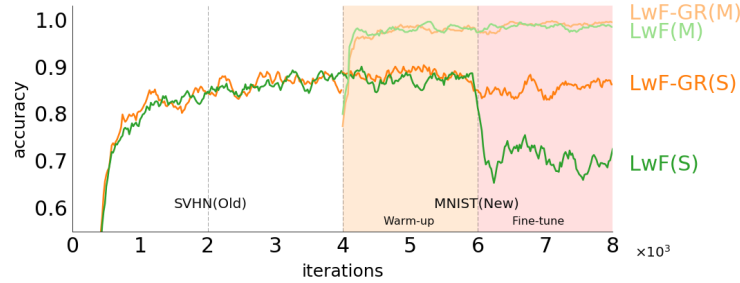

Figure 5: Performance of LwF and LwF augmented with generative replay (LwF-GR) on classifying samples from each domain. The networks were trained on SVHN then on MNIST database. Test accuracy on SVHN classification task (thick curves) dropped when the shared parameters were fine-tuned, but generative replay greatly tempered the loss (orange). Both networks achieved high accuracy on MNIST classification (dim curves).

network (green). However, with generative replay, the network maintains most of the past knowledge (orange).

## 4.3 Learning new classes

To illustrate that generative replay can recollect the past knowledge even when the inputs and targets are highly biased between the tasks, we propose a new experiment in which the network is sequentially trained on disjoint data. In particular, we assume a situation where the agent can access examples of only a few classes at a time. The agent eventually has to correctly classify examples from all classes after being sequentially trained on mutually exclusive subsets of classes. We tested the networks on MNIST handwritten digit database.

Note that training the artificial neural networks independently on classes is difficult in standard settings, as the network responses may change to match the new target distribution. Hence replaying inputs and outputs that represent former input and target distributions is necessary to train a balanced network. We thus compare the variants described earlier in this section from the perspective of whether the input and target distributions of cumulative real data is recovered. For *ER* and *GR* models, both the input and target distributions represent cumulative distribution. *Noise* model maintains cumulative target distributions, but the input distribution only mirrors current distribution. *None* model has current distribution for both.

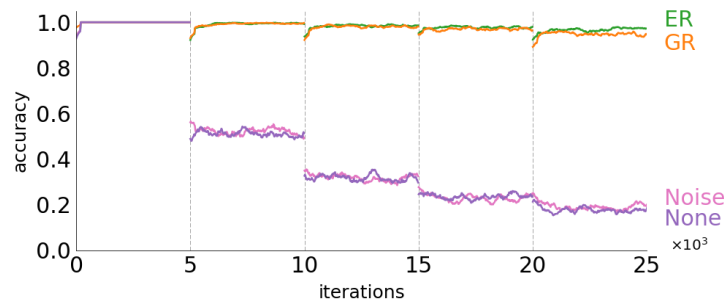

Figure 6: The models were sequentially trained on 5 tasks where each task is defined to classify MNIST images belong to 2 out of 10 labels. In this case, the networks are given with examples of 0 and 1 during the first task, 2 and 3 for the second, and in the same manner. Only our networks achieved test performance close to the upper bound.

In Figure 6, we divided MNIST dataset into 5 disjoint subsets, each of which contains samples from only 2 classes. When the networks are sequentially trained on the subsets, we observed that a naively trained classifier completely forgot previous classes and only learned the new subset of data (purple). Recovering only the past output distribution without a meaningful input distribution did not help retaining knowledge, as evidenced by the model with a noise generator (pink). When both the input

and output distributions are reconstructed, generative replay evoked previously learnt classes, and the model was able to discriminate all encountered classes (orange).

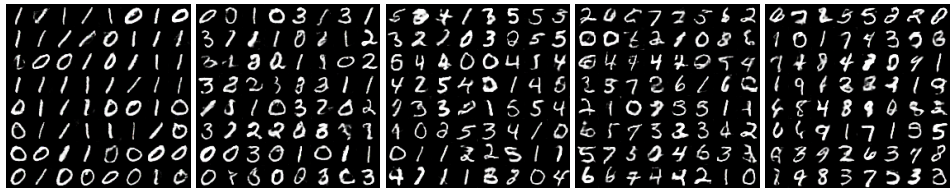

Figure 7: Generated samples from trained generator after the task 1, 2, 3, 4, and 5. The generator is trained to reproduce cumulative data distribution.

Because we assume that the past data are completely discarded, we trained the generator to mimic both current inputs and the generated samples from the previous generator. The generator thus reproduces cumulative input distribution of all encountered examples so far. As shown in Figure 7, generated samples from trained generator include examples equally from encountered classes.

## 5  Discussion

We introduce deep generative replay framework, which allows sequential learning on multiple tasks by generating and rehearsing fake data that mimics former training examples. The trained scholar model comprising a generator and a solver serves as a knowledge base of a task. Although we described a cascade of knowledge transfer between a sequence of scholar models, a little change in formulation proposes a solution to other topically relevant problems. For instance, if the previous scholar model is just a past copy of the same network, it can learn multiple tasks without explicitly partitioning the training procedure.

As comparable approaches, regularization methods such as EWC and careful training the shared parameters as in LwF have shown that catastrophic forgetting could be alleviated by protecting former knowledge of the network. However, regularization approaches constrain the network with additional loss terms for protecting weights, so they potentially suffer from the tradeoff between the performances on new and old tasks. To guarantee good performances on both tasks, one should train on a huge network that is much larger than normally needed. Also, the network has to maintain the same structure throughout all tasks when the constraint is given specific to each parameter as in EWC. Drawbacks of LwF framework are also twofold: the performance highly depends on the relevance of the tasks, and the training time for one task linearly increases with the number of former tasks.

The deep generative replay mechanism benefits from the fact that it maintains the former knowledge solely with input-target pairs produced from the saved networks, so it allows ease of balancing the former and new task performances and flexible knowledge transfer. Most importantly, the network is jointly optimized towards task objectives, hence guaranteed to achieve the full performance when the former input spaces are recovered by the generator. One defect of the generative replay framework is that the efficacy of the algorithm heavily depends on the quality of the generator. Indeed, we observed some performance loss while training the model on SVHN dataset within same setting employed in section 4.3. Detailed analysis is provided in supplementary materials.

We acknowledge that EWC, LwF, and ours are not completely exclusive, as they contribute to memory retention at different levels. Nevertheless, each method poses some constraints on training procedure or network configurations, and there is no straightforward mixture of any two frameworks. We believe a good mix of the three frameworks would give a better solution to the chronic problem in continual learning.

Future works of generative replay may extend to reinforcement learning domain or the form of continuously evolving network that maintains knowledge from past copy of the self. Also, we expect the improvements in training deep generative models would directly aid the performance of generative replay framework on more complex domains.

## Acknowledgement

We would like to thank Hyunsoo Kim, Risto Vuorio, Joon Hyuk Yang, Junsik Kim and our reviewers for their valuable feedback and discussion that greatly assisted this research.

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
