[Reviews · NeurIPS 2017]

Reviewer 1



Quality: The paper is technically sound. Extensive comparisons between the proposed approach and other baseline approaches are being made. Clarity: The paper is well-organized. However, insufficient details are provided on the architectures of the discriminator, generator and the used optimizer. I would be surprised if the cited papers refer to the hippocampus as a short-term memory system since the latter typically refers to working memroy. I assume they all refer to hippocampus as a system for long-term memory consolidation. I advice to adapt accordingly. A bit more attention should be paid to English writing style: primate brain’s the most apparent distinction inputs are coincided with Systems(CLS) Recent evidences networks(GANs) A branch of works line 114: real and outputs 1/2 everywhere => unclear; please reformulate line 124: real-like samples => realistic; real-life (?) in both function past copy of a self. Sequence of scholar models were favorable than other two methods can recovers real input space Old Tasks Performance good performances with in same setting employed in section 4.3. 
 Originality: The proposed approach to solve catastrophic forgetting is new (to my knowledge) yet also straightforward, replacing experience replay with a replay mechanism based on a generative model. The authors do show convincingly that their approach performs much better than other approaches, approximating exact replay. A worry is that the generative model essentially stores the experiences which would explain why its performance is almost identical to that of experience replay. I think the authors need to provide more insight into the how performance scales with data size and show that the generative model becomes more and more memory efficient as the number of processed training samples increases. In other words: the generative model only makes sense if its memory footprint is much lower than that of just storing the examples. I am sure this would be the case but it is important and insightful to make this explicit. Significance: The results are important given the current interest in life-long learning. While the idea is expected in the light of current advances, the authors give a convincing demonstration of the usefulness of this approach. If the authors can improve the manuscript based on suggested modifications then I believe the paper would be a valuable contribution.

Reviewer 2



This paper introduces a model for performing continual/transfer learning by using a generative network/solver pair (old scholar) to continually replay fake data based on previously learned information for the new scholar to learn. The generative network is trained using the GAN framework. The entire model is trained on a sequence of tasks, with the n'th scholar training on real data from the current task Tn as well as fake data generated by the previous scholar n-1. They tested on MNIST and SVHN, examining transfer from one dataset to the other as well as from fewer classes to more classes in MNIST. This is a great idea, and overall the results are impressive. However the paper is a bit confusingly written and the model needs to be explained in more detail. For instance, what is the ratio of real to replayed samples that new scholars learn on? Does this hyperparameter need to be tuned for different tasks? The paper also needs some proofing, as many sentences had grammatical errors, such as the caption for Table 2: "Our generative replay algorithm is favorable than other two methods in that it is equivalent to joint training on accumulated real data as long as the trained generator can recovers real input space." I like that both a ceiling control (Exact Replay) as well as a random control (Noise) were both provided, as this gave a great way to assess the performance of their model in absolute terms. However, a baseline comparison with another SOTA model for continual learning such as EWC or LwF would have been even better. This lack of a baseline is I think the biggest weakness of this submission. I am appreciative of the fact that performance comparisons with other models are not always interpretable, given different constraints on capacity, flexibility, etc, that the authors outline in Table 2, but some attempt to quantify these differences would have been appreciated, rather than asking the reader to simply take these relative advantages/disadvantages on faith.